# Distinct roles of NMDA receptors at different stages of granule cell development in the adult brain

Yangling Mu[1,2†], Chunmei Zhao[3†], Nicolas Toni[4], Jun Yao[5], Fred H Gage[3*]

[1]Department of Physiology, School of Basic Medicine, Huazhong University of Science and Technology, Wuhan, China; [2]Institute of Brain Research, Collaborative Innovation Center for Brain Science, Huazhong University of Science and Technology, Wuhan, China; [3]Laboratory of Genetics, Salk Institute for Biological Studies, La Jolla, United States; [4]Department of Fundamental Neurosciences, University of Lausanne, Lausanne, Switzerland; [5]State Key Laboratory of Biomembrane and Membrane Biotechnology, Tsinghua-Peking Center for Life Sciences, School of Life Sciences, Tsinghua University, Beijing, China

**Abstract** NMDA receptor (NMDAR)-dependent forms of synaptic plasticity are thought to underlie the assembly of developing neuronal circuits and to play a crucial role in learning and memory. It remains unclear how NMDAR might contribute to the wiring of adult-born granule cells (GCs). Here we demonstrate that nascent GCs lacking NMDARs but rescued from apoptosis by overexpressing the pro-survival protein Bcl2 were deficient in spine formation. Insufficient spinogenesis might be a general cause of cell death restricted within the NMDAR-dependent critical time window for GC survival. NMDAR loss also led to enhanced mushroom spine formation and synaptic AMPAR activity throughout the development of newborn GCs. Moreover, similar elevated synapse maturation in the absence of NMDARs was observed in neonate-generated GCs and CA1 pyramidal neurons. Together, these data suggest that NMDAR operates as a molecular monitor for controlling the activity-dependent establishment and maturation rate of synaptic connections between newborn neurons and others.

*For correspondence: gage@salk.edu

†These authors contributed equally to this work

Competing interests: The authors declare that no competing interests exist.

## Introduction

New neurons are continuously generated in the hippocampus of the adult mammalian brain. These neurons become granule cells (GCs), the principal neurons in the dentate gyrus (DG) of the hippocampus, and they functionally integrate into the hippocampal circuitry (*van Praag et al., 2002*; *Toni et al., 2008*). This extreme form of structural remodeling, similar to many other forms of experience-dependent plasticity, requires activation of NMDA receptors (NMDARs) (*Platel and Kelsch, 2013*). Blockade of NMDARs rapidly increases the proliferation of neural precursor cells, whereas stimulation of NMDARs promotes neuronal fate specification (*Cameron et al., 1995*; *Deisseroth et al., 2004*). Moreover, deletion of the NMDAR subunit NR1 reduces the survival rate of adult-born GCs (*Tashiro et al., 2006a*). Other than these studies, the role of NMDAR in circuit assembly during new neuron development has so far received little attention, even though NMDAR has been primarily known for its involvement in synapse organization and synaptic plasticity (*Constantine-Paton, 1990*; *Malenka and Nicoll, 1993*).

The wiring of new neurons in mature circuits involves a coordinated series of events, from the initial cell contact to the final maturation of functional synapses. Not only the cells themselves but also the connections between newly recruited members and their old counterparts are survivors of a selection

**eLife digest** The brain contains billions of cells called neurons. Although most neurons have already formed by the time we are born, part of the brain called the hippocampus produces new neurons throughout our life. These new neurons are thought to be important for learning and forming new memories.

Neurons send signals to each other across connections called synapses. Small protrusions called spines stick out of the neuron and each tends to have one synapse that receives a signal from another neuron. Via these connections, the neurons are organized into networks and circuits that control how different parts of the brain work. Therefore, once new neurons are made, they also need to be connected to the correct neurons.

The NMDA receptor is found in the surface of neurons, and mutated neurons that lack this receptor often die shortly after birth. The NMDA receptor is also known to be important for organizing synapses. Exactly how NMDA receptors help new neurons to survive and integrate into circuits has not been investigated in detail. Mu, Zhao et al. now address this issue by using mice in which a gene called NR1, which produces one of the proteins that makes up the NMDA receptor, can be deleted at specific stages of neuron development.

Analyzing brain slices from the mice showed that deleting NR1 from newly-formed neurons caused them to die within two or three weeks. When these neurons were forced to survive, they had fewer spines than normal.

By contrast, deleting NR1 from neurons that has already survived for longer than four weeks did not alter how many spines the neurons had. Instead, the synapses on the spines worked better. Mu, Zhao et al. therefore suggest that NMDA receptors have different roles at different stages of a neuron's development. Initially, NMDA receptors help the neurons to survive and form spines. The receptors then help to ensure the spines become the correct size, and enable the neurons to connect into the right neural circuits by helping to control the strength of synapses.

Mu, Zhao et al. theorize that the mere presence of NMDA receptors suppresses spine maturation. Furthermore, this inhibitory effect is only released when the NMDA receptor is activated, or when the NMDA receptor is absent due to the deletion of the NR1 gene. Further studies will be needed to validate this hypothesis.

process depending upon neuronal activity patterns. The vast majority of excitatory inputs are received by small bulbous protrusions residing on the dendrites of glutamatergic neurons, called dendritic spines. In newborn GCs, dendritic spines first appear around 16 days after neuronal birth. The spine density increases sharply before cells reach 4 weeks of age and continues to increase at a slower pace until reaching a plateau at 8 weeks (*Zhao et al., 2006*). Notably, the NMDAR-dependent survival/ death of adult-born GCs is restricted to the time window of 2–3 weeks after neuronal birth, shortly after formation of the first dendritic spines (*Tashiro et al., 2006a*). This temporal overlap suggests that the survival of newborn GCs may be related to the state of spinogenesis or synaptogenesis. Indeed, cell death can be induced by non-innervation at the peak of synaptogenesis during embryonic and postnatal development (*Naruse and Keino, 1995*). Since activation of NMDARs has been shown to support new spine formation (*Maletic-Savatic et al., 1999*; *Kwon and Sabatini, 2011*), we hypothesize that NMDAR is required for initial spine gain on dendrites of newborn GCs and that insufficient spine growth may be the underlying cause of the cell death associated with the genetic deletion of the NMDAR subunit NR1 in newborn GCs.

There is a positive correlation between spine volume and the number of AMPA (α-amino-3-hydroxy-5-methyl-4-isoxazole propionic acid) receptors (AMPARs) or, more generally, the synaptic strength (*Matsuzaki et al., 2001*), supporting the model that spine outgrowth and enlargement are tightly coupled to the formation and maturation of glutamatergic synapses (*Zito et al., 2009*). Subject to activity-dependent modifications, spines are highly dynamic in their number, shape and size. Long-term potentiation (LTP) and long-term depression (LTD) at mature synapses, expressed by synaptic insertion and removal of AMPARs, respectively, are associated with NMDAR-dependent enlargement and shrinkage of dendritic spines (*Yuste and Bonhoeffer, 2001*; *Matsuzaki et al., 2004*; *Zhou et al., 2004*). Stimuli that induce LTP or LTD may also result in rapid outgrowth or loss of spines, and these

changes can be prevented by NMDAR blockers (*Engert and Bonhoeffer, 1999*; *Toni et al., 1999*; *Nagerl et al., 2004*). The structural plasticity of spines is thought to complement functional plasticity (e.g., LTP and LTD) and play a central role in learning and memory in mature animals (*Bourne and Harris, 2007*). In contrast, emerging evidence suggests that, during early postnatal development, AMPARs can be delivered to spines independently of NMDAR signaling. NMDARs actually restrict AMPAR trafficking to the postsynaptic density and limit synapse maturation (*Ultanir et al., 2007*; *Adesnik et al., 2008*; *Gray et al., 2011*). Due to variables in experimental designs, the observed opposite effects of NMDARs on AMPAR recruitment in developing vs mature neurons need further examination. Adult-born neurons undergo a long process of maturation resembling embryonic development (*Esposito et al., 2005*); however, adult neurogenesis happens in mature circuits that differ substantially from the developing brain. We aimed to characterize NMDAR functions during integration of new GCs into the circuit at distinct stages of their development using the tool of single-cell gene deletion and labeling.

To address the above issues, we performed an analysis of spine morphogenesis in immature NR1 knockout (KO) GCs that either survived naturally or were rescued by the apoptosis regulator Bcl-2. Both groups of cells were found to be deficient in spine formation. In parallel, we observed an elevation in mushroom spine density and in synaptic AMPAR activity in the absence of NR1. Furthermore, NMDAR loss initiated at a later stage of GC development, similar to NR1 KO in fully mature GCs or CA1 pyramidal neurons, resulted in enhanced functional synapses without affecting total spine numbers. Thus, NMDAR appears to play two distinct roles during GC development. First, it promotes the initial spine formation and its presence is required for the survival of immature GCs. Second, the receptors monitor spine enlargement and the recruitment of AMPAR once spines are formed. Both aspects of NMDAR function contribute to experience-driven construction of circuits formed by new neurons, even though NMDAR signaling might make less of a contribution to the control of overall spine density upon neuronal maturation.

## Results

### NMDAR loss in newborn GCs leads to decreased spine density and increased mushroom spine density

To determine the role of NMDAR in circuit assembly of new neurons in the adult brain, we first examined the morphology of NR1 KO GCs at 4 weeks of age. A retrovirus encoding Cre recombinase and GFP was developed for inducible knockout of the floxed *Grin1* alleles in adult mice (rv GFP-ires-cre; *Figure 1—figure supplement 1*). When tested in the ROSA-lacZ reporter mice, rv GFP-ires-cre induced recombination in 97% of GFP+ cells at 6 days post infection (dpi) and in all GFP+ cells at 14 and 28 dpi. We then injected rv GFP-ires-cre together with a control retroviral vector expressing mCherry only (rv CAG-mCherry) into the *Grin1* floxed mice (*Tsien et al., 1996*; *Tashiro et al., 2006a*) (*Figure 1A*). To assess NMDAR activity in virus-transduced cells and confirm the cell-specific knockout of the *Grin1* gene via Cre/loxP recombination, we performed perforated whole-cell patch-clamp recordings at a holding potential of −70 mV and +40 mV to monitor synaptic responses mediated by AMPA and NMDARs, respectively. In control adult-born GCs (mcherry+GFP−) at 28 dpi, both AMPA and NMDA currents could be readily evoked by perforant path stimulation (*Figure 1B*). In contrast, there was only a DNQX-sensitive AMPA component in age-matched GFP+ neurons infected by rv GFP-ires-cre (*Figure 1B*), suggesting that Cre-mediated recombination successfully removed the floxed *Grin1* gene fragment from the mouse genomic DNA.

Because the fluorescent signal produced by rv CAG-mCherry labeling was not sufficient for optimal image acquisition and analysis of dendritic spines, we then injected the control CAG-GFP or GFP-ires-cre retrovirus into *Grin1^{f/f}* mice to compare the morphology of NR1 wild-type (WT) and KO cells. There was no obvious difference in overall cell morphology between NR1 KO and WT cells (*Figure 1C,D*). Dendritic tracing with the ICL TRACE (http://synapses.clm.utexas.edu/tools/trace/trace.stm) showed that WT and KO cells were similar in both dendritic length (WT: 601.8 ± 36.6, n = 47 frames, KO: 515.1 ± 29.3, n = 38 frames, p = 0.08) and branching points (WT: 5.92 ± 0.33, KO: 5.68 ± 0.37, p = 0.64). These data indicate that gross development of dendrites does not require NMDARs. However, detailed analyses of the dendritic segments in the outer third of the molecular layer revealed significant differences between WT and KO cells (*Figure 1E*). According to the criteria described by Harris and Yuste (*Harris et al., 1992*; *Parnass et al., 2000*), all dendritic protrusions were classified

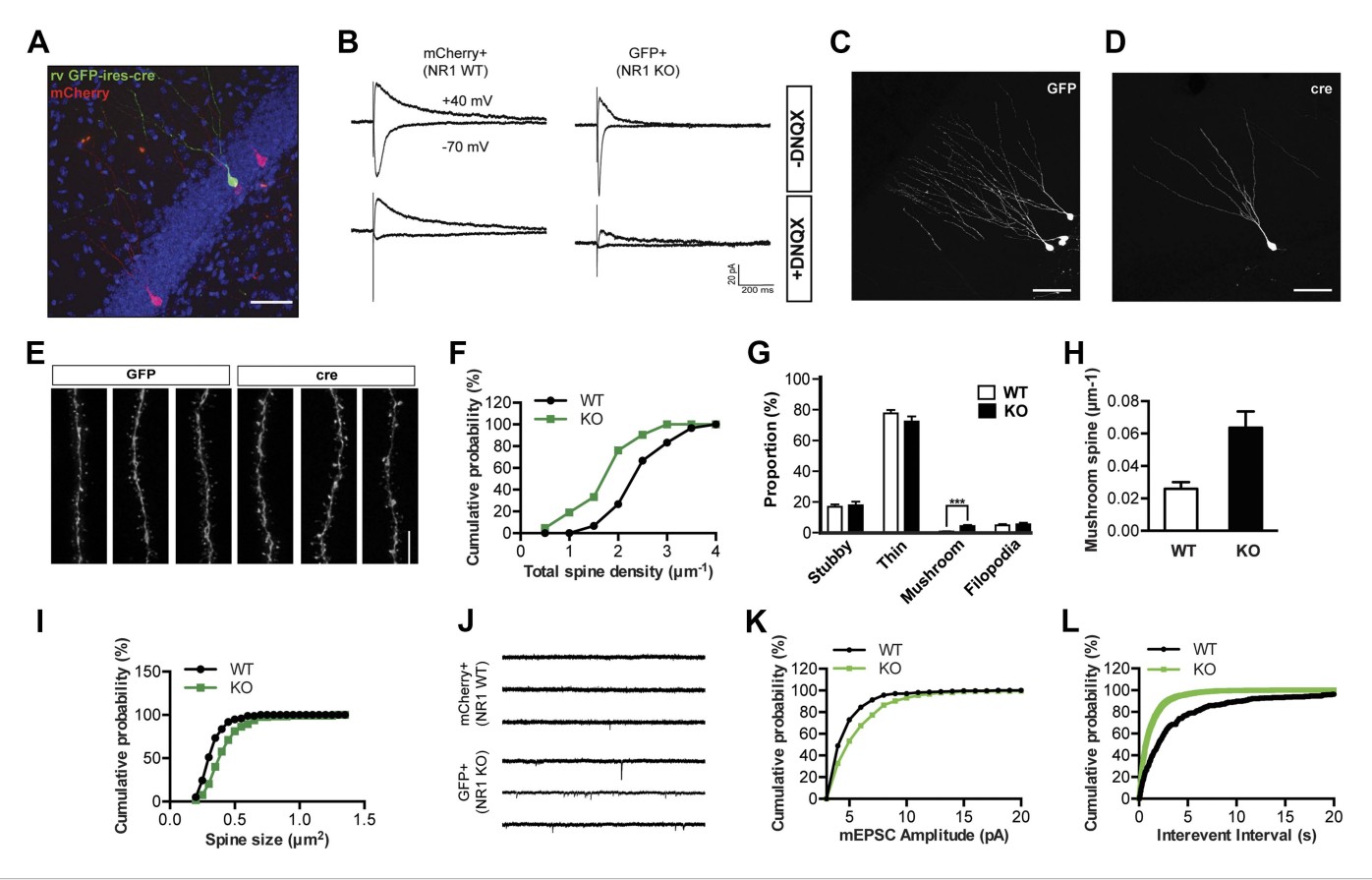

**Figure 1**. NR1 KO cells display decreased spine growth but enhanced spine maturation and AMPAR activity at 4 weeks of age. (**A**) Co-injection of rv CAG-mcherry (red) and CAG-GFP-ires-cre (green) for the simultaneous labeling of wild-type (WT) and NR1 KO newborn granule cells (GCs). (**B**) Left: mCherry+ newborn GCs respond to perforant path stimulation in the absence (upper panel) and presence (lower panel) of the AMPAR antagonist DNQX. Right: GFP+ Cre-expressing newborn GCs respond to perforant path stimulation in the absence (upper panel) but not in the presence (lower panel) of DNQX. (**C**, **D**) Representative images of WT (**C**) and NR1 KO (**D**) newborn GCs at 4 weeks of age. (**E**) Representative images of dendritic processes of newborn WT (GFP) and NR1 KO (cre) GCs in the outer molecular layer. (**F**) Total spine density is decreased in NR1 KO newborn GCs. (**G**) Comparison of the percentage of each spine type relative to total spine numbers in adult-born WT and NR1 KO GCs. (**H**) Mushroom spine density is increased in NR1 KO newborn GCs. (**I**) Cumulative plot of spine size in NR1 WT and KO GCs. (**J**) Representative traces of AMPAR-mediated miniature excitatory postsynaptic currents (mEPSCs) in mCherry+ WT and GFP+ NR1 KO newborn GCs. (**K**, **L**) Quantitative analysis of mEPSCs by amplitude (**K**) and frequency (**L**).

The following figure supplement is available for figure 1:

**Figure supplement 1**. Retrovirus rv CAG GFP-ires-cre was delivered to the dentate gyrus (DG) of ROSA-lacZ mice and the recombination efficiency was examined by the expression of β-gal (red) in Cre-expressing cells (GFP+, green).

into four categories: filopodia, stubby, thin and mushroom spines. Briefly, 'stubby' were neckless spines whose head diameters were about equal to their lengths. Spines were defined as 'filopodia' if they were long, thin and did not have a head. In contrast, 'thin' spines had long, thin necks and obvious heads. 'Mushroom' spines were similar to 'thin' spines in shape, but with larger heads (see 'Materials and methods'). As shown in *Figure 1F*, total spine density was significantly decreased in NR1 KO cells (WT: 2.14 ± 0.08, n = 59 frames, KO: 1.53 ± 0.09, n = 37 frames, p < 0.0001). Surprisingly, the percentage of mushroom type relative to total spine numbers was drastically enhanced, whereas the other spine types remained unchanged (p < 0.0001; *Figure 1G*). In line with this observation, mushroom spine density was increased by more than twofold in NR1 KO cells (WT: 0.026 ± 0.004, KO: 0.064 ± 0.010, p = 0.0001; *Figure 1H*). A cumulative probability graph of the size of all measured spines showed that the spine head area in NR1 KO cells was bigger than that in control cells (p < 0.0001, Kolmogorov–Smirnov test; *Figure 1I*).

The size of the spine head has been positively correlated with synaptic AMPAR level (*Matsuzaki et al., 2001*). Since NR1 KO cells displayed increased mushroom spines, we postulated that NMDAR KO cells might have more synaptic AMPAR activity. Therefore, we measured AMPAR-mediated miniature excitatory postsynaptic currents (mEPSCs). While mEPSCs were infrequent in mCherry+GFP− NR1 WT cells, they were evident in GFP+ NR1 KO cells (*Figure 1J*). Both the amplitude and frequency of mEPSCs were significantly increased in neurons lacking NR1 as compared to control cells (WT: n = 5 cells, KO: n = 6 cells, p < 0.001, Kolmogorov–Smirnov test; *Figure 1K,L*). These data confirmed that NR1 KO cells had enhanced functional glutamatergic synapses. Utilizing serial immuno-electron microscopy for GFP, we consistently observed that GFP+ dendritic spines were associated with GFP− axon terminals containing presynaptic vesicles (*Figure 2A*), verifying that newborn NR1-null GCs could form normal synapses. We also noted that all stubby, thin and mushroom spines were asymmetric and presumably excitatory (*Figure 2A*). While the axo-dendritic synapses located on dendritic shafts had a typical morphology of symmetric or GABAergic synapses (*Figure 2A*), they could be initially excitatory in newly born GCs due to the high intracellular chloride concentration (*Ge et al., 2006*). Total head volumes of randomly selected spines averaged $0.035 \pm 0.005$ μm³ (n = 130 spines) and $0.078 \pm 0.014$ μm³ (n = 74 spines) in WT and NR1 KO GCs, respectively, indicating a significant difference between these two groups (p = 0.0008; *Figure 2B*). Furthermore, we found that this difference was mainly due to increased volume of big or mushroom spines in NR1 KO neurons, whereas the volumes of small spines (presumably filopodia or thin) were roughly the same across groups. These results suggest that changes in the amount of depolarization-induced $Ca^{2+}$ influx

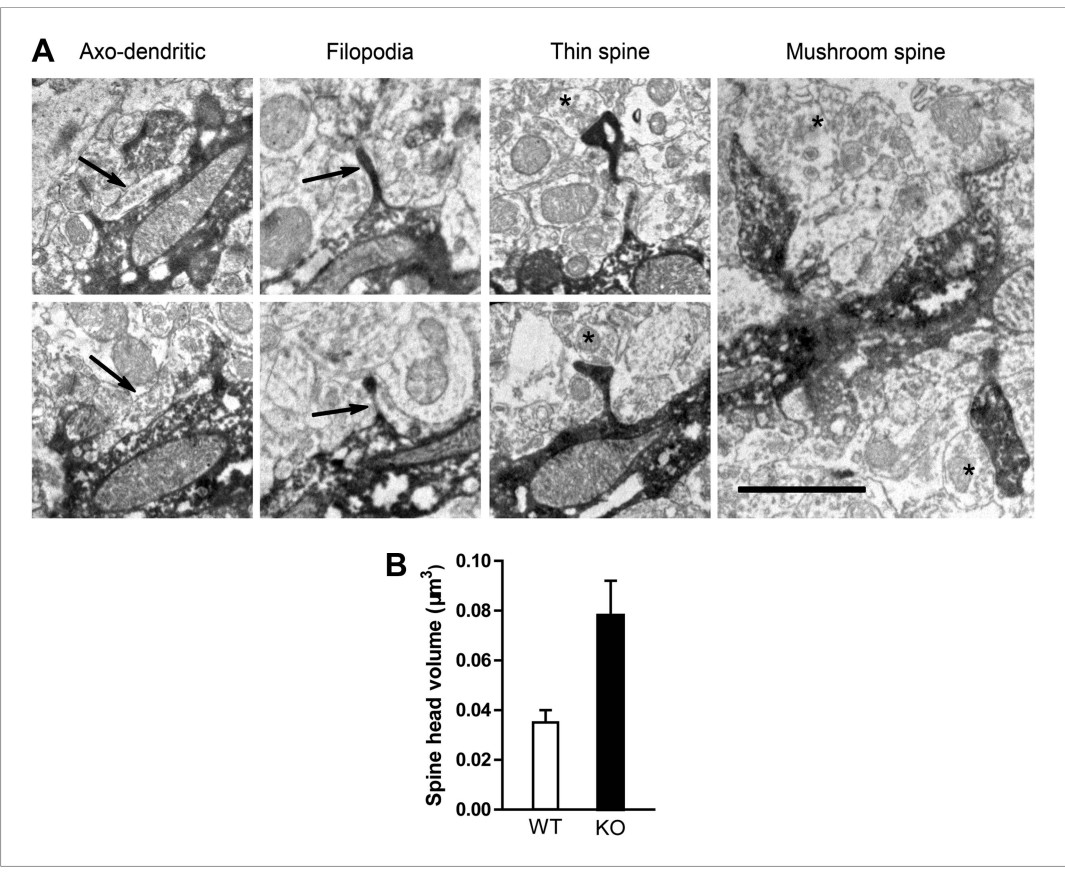

**Figure 2**. Electron microscopic description of dendritic spines. (**A**) Electron micrograph illustrating dendrites of newborn NR1 KO neurons. Panels show examples of symmetric axo-dendritic synapses (left panels, arrows), filopodia (middle left panels), thin spines (middle right panels) and mushroom spines (right panel). Darkly immunolabeled GFP+ dendritic spines are each contacted by GFP− axon terminals (asterisks) containing numerous presynaptic vesicles. Scale bars: 1 μm. (**B**) Comparison of total spine volumes in NR1 KO and WT cells.

may affect the survival of nascent neurons, although the role of a possible re-distribution of $Ca^{2+}$ entry through different spine categories cannot be excluded.

## NMDAR is required for spine growth in newborn GCs

Given that most NR1 KO GCs die before reaching 4 weeks of age (*Tashiro et al., 2006a*), it is questionable whether the observed abnormal spine morphogenesis in the few surviving NR1 KO cells represents a general phenomenon in NR1-null neurons. To clarify this issue, we engineered a new retroviral vector expressing a fusion protein of the pro-survival gene Bcl2 and GFP in addition to Cre recombinase (rv GFPBcl2-ires-cre; *Figure 3—figure supplement 1*) to prevent NR1 KO cells from dying. The logic behind this set of experiments is: NR1 deletion in new GCs results in certain defects that eventually lead to cell death. Since Bcl2 is an important anti-apoptotic protein (*Tsujimoto et al., 1984*; *Cleary et al., 1986*), it may suppress apoptosis and drive NR1 KO cells to survive even though they have deficiencies. Therefore, neurons rescued by Bcl2 expression should exhibit features distinct from those surviving normally, and these differences presumably reflect defects caused by NR1 loss. We first tested the virus efficiency of cell death regulation in WT C57Bl/6 mice. In line with prior studies (*Dayer et al., 2003*; *Tashiro et al., 2006b*), within 28 dpi of rv CAG-GFP, we found a significant decrease in the quantity of GFP+ GCs in C57Bl/6 mice (GFP, relative cell number to that of 14 dpi, 14 dpi: $1.00 \pm 0.11$, n = 5 mice, 28 dpi: $0.30 \pm 0.22$, n = 4 mice, p = 0.019; *Figure 3A*). In contrast, GCs infected by rv GFPBcl2-ires-cre did not die (GFPBcl2icre, 14 dpi: $1.00 \pm 0.29$, n = 6 mice, 28 dpi: $1.00 \pm 0.14$, n = 6 mice; *Figure 3A*). More importantly, the dramatic death of new neurons observed in *Grin1^{f/f}* mice injected with rv GFP-ires-cre was not found in *Grin1*-floxed animals injected with rv GFPBcl2-ires-cre (GFPicre, 7 dpi: $1.00 \pm 0.37$, n = 5 mice, 21 dpi: $0.05 \pm 0.03$, n = 5 mice, p = 0.032; GFPBcl2icre, 7 dpi: $1.00 \pm 0.31$, n = 5 mice, 21 dpi: $1.31 \pm 0.39$, n = 5 mice, p = 0.5; *Figure 3B*), suggesting that the expression of Bcl2 did prevent the death of NR1 KO cells.

Next we assessed the morphology of NR1 KO cells that were rescued by Bcl2 expression. Because the fluorescent signal from GFPBcl2 fusion protein was not strong enough for morphological analyses, rv GFPBcl2-ires-cre was injected together with rv CAG-GFP. In this case, we identified rv GFPBcl2-ires-cre targeted cells by Cre immunostaining and imaged Cre+GFP+ cells when the newborn cells were 4 weeks old (*Figure 3C*). GFP-labeled GCs from animals injected with pure rv CAG-GFP or rv GFP-ires-cre were imaged in parallel for control purposes. In C57Bl/6 mice, Bcl2-expressing cells displayed decreased dendritic length (GFP: $378.2 \pm 32.0$, n = 58 frames, cre: $351.5 \pm 45.6$, n = 33 frames, Bcl2: $283.8 \pm 22.0$, n = 93 frames, p = 0.013 Bcl2 vs GFP; *Figure 3D*). The number of dendritic branching points was not significantly different between the samples (GFP: $5.2 \pm 0.3$, cre: $4.9 \pm 1.5$, Bcl2: $4.4 \pm 0.2$, p = 0.056 Bcl2 vs GFP, *Figure 3E*). Similarly, in *Grin1^{f/f}* mice, Bcl2-expressing cells displayed a significant decrease in dendritic length (GFP: $342.4 \pm 32.9$, n = 55 frames, cre: $302.7 \pm 22.8$, n = 85 frames, Bcl2: $256.0 \pm 18.0$, n = 133 frames, p = 0.014 Bcl2 vs GFP; *Figure 3F*) but not in branching points (GFP: $5.4 \pm 0.4$, cre: $5.1 \pm 0.3$, Bcl2: $4.7 \pm 0.3$, p = 0.14 Bcl2 vs GFP, *Figure 3G*). Since NR1 KO neurons did not display impairment of dendrite length and complexity (*Figure 1C,D*), we infer from these data that Bcl2 expression promoted survival of a mixed population of nascent GCs, including both WT and NR1 KO cells.

Detailed analyses of the dendritic processes in the outer molecular layer showed that Bcl2-expressing cells in C57Bl/6 mice had much lower total spine density (GFP: $2.13 \pm 0.11$, n = 28 frames, cre: $2.22 \pm 0.11$, n = 17 frames, Bcl2: $1.39 \pm 0.14$, n = 31 frames, p = 0.0001 Bcl2 vs GFP; *Figure 3H,I*). A closer examination of the data identified a unique population of dendritic branches with low spine density (<1.5 spines/μm) in the Bcl2-expressing group but not in samples expressing GFP or Cre alone, although the cumulative fractions of spine counts in these groups were not statistically different (p > 0.1, Kolmogorov–Smirnov test; *Figure 3J*). However, neither the mushroom spine density (GFP: $0.015 \pm 0.006$, cre: $0.013 \pm 0.004$, Bcl2: $0.017 \pm 0.006$) nor the percentage of mushroom spines (GFP: $0.80 \pm 0.27$, cre: $1.06 \pm 0.30$, Bcl2: $1.08 \pm 0.34$) relative to total number of protrusions showed any difference between samples in C57Bl/6 mice (*Figure 3K,L*). These results suggest that WT adult-born GCs selected for death had fewer spines, specifically fewer thin spines (p < 0.05; *Figure 3L*), as compared to those destined for survival. In *Grin1^{f/f}* mice, deletion of NMDAR in newborn GCs led to a decrease in total spine density, and the spine density was even lower in Bcl2-rescued cells (GFP: $1.96 \pm 0.06$, n = 68 frames, cre: $1.47 \pm 0.11$, n = 35 frames, Bcl2: $1.06 \pm 0.08$, n = 45 frames, p = 0.0054 cre vs Bcl2; *Figure 3M,N*), whereas the cumulative distribution of total

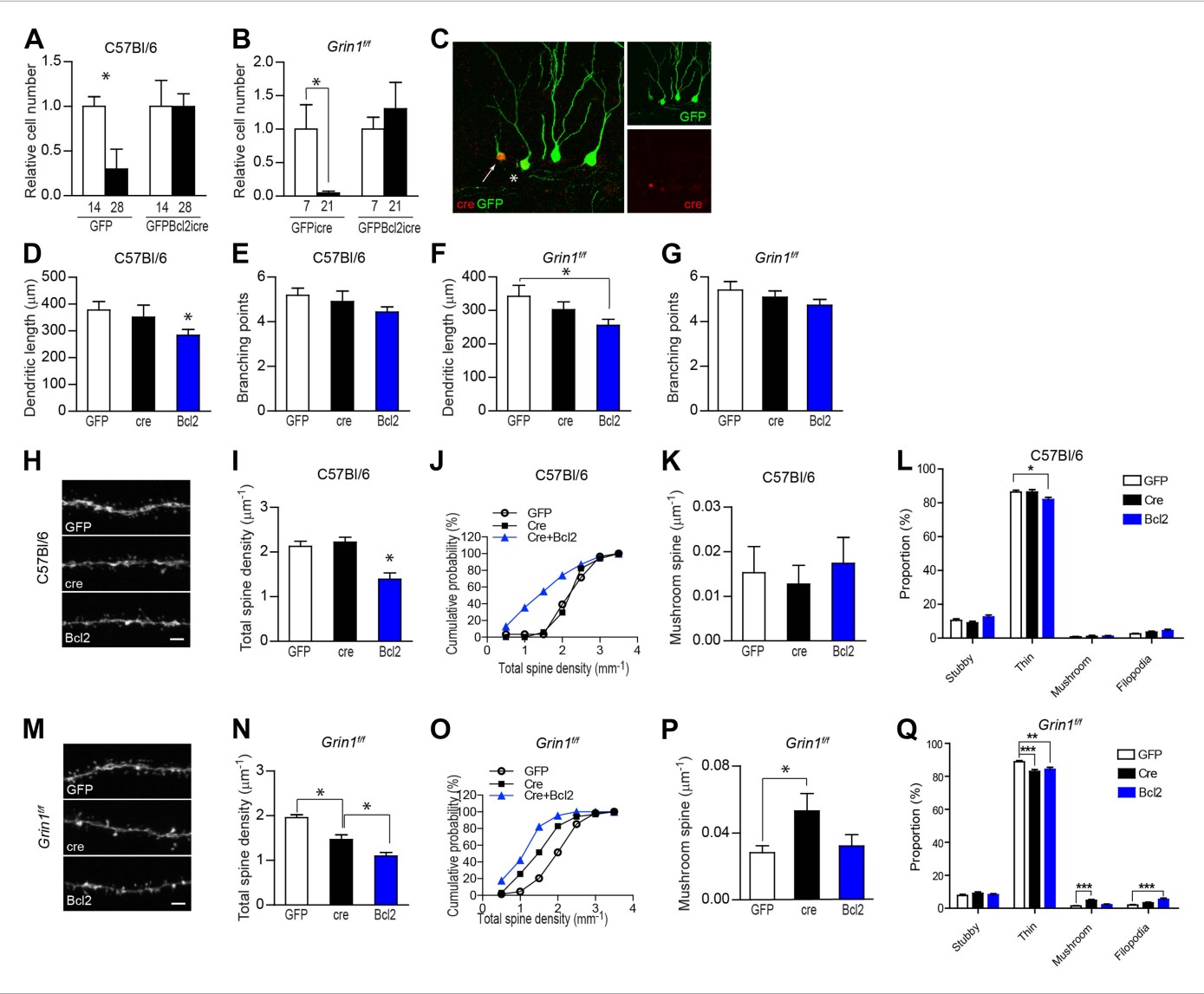

**Figure 3**. NR1 KO cells rescued by expression of the pro-survival gene Bcl2 had an unusually low spine density. (**A**) Bcl2 expression was able to rescue newborn GCs from naturally occurring cell death in C57Bl/6 mice. (**B**) Bcl2 expression was able to rescue NR1 KO newborn GCs from cell death. (**C**) For morphological analysis of GFPBcl2-ires-cre-targeted cells, rv GFPBcl2-ires-cre was co-injected with rv GFP. Bcl2-ires-cre-targeted cells were identified by immunohistochemistry using an antibody against the cre recombinase. The arrow and asterisk show GCs transfected by both rv GFPBcl2-ires-cre and rv GFP. Insets on the right side represent GFP− (green) and Cre-expressing (red) GCs, respectively. (**D–G**) Analysis of dendritic growth of newborn GCs at 28 dpi. Bcl2-rescued newborn GCs had a decreased total dendritic length in both C57Bl/6 and *Grin1f/f* mice (**D**, **F**), whereas branching points were not changed (**E**, **G**). (**H**) Representative images of dendritic processes in the outer molecular layer of newborn GCs at 28 dpi in C57Bl/6 mice labeled by rv GFP, GFP-ires-cre and GFPBcl2-ires-cre. (**I**, **J**) Total spine density was similar in GFP and GFP-ires-cre targeted newborn GCs but significantly decreased in GFPBcl2-ires-cre targeted newborn GCs. (**K**) The density of mushroom spines did not change significantly in newborn GCs targeted by the three retroviruses. (**L**) Comparison of the percentage of each spine type relative to total spine numbers in new GCs of C57Bl/6 mice infected by the three retroviruses. (**M**) Representative images of dendritic processes in the outer molecular layer of newborn GCs at 28 dpi in *Grin1f/f* mice labeled by rv GFP, GFP-ires-cre and GFPBcl2-ires-cre. (**N**, **O**) Total spine density was significantly lower in surviving NR1 KO cells (Cre) and further decreased in Bcl2-rescued cells. (**P**) Mushroom spine density was increased in NR1 KO cells (Cre) but not in Bcl2-rescued cells. (**Q**) Comparison of the percentage of each spine type relative to total spine numbers in new GCs of *Grin1f/f* mice infected by the three retroviruses. Scale bars: 20 μm (**C**) and 2 μm (**H**, **M**).

The following figure supplement is available for figure 3:

**Figure supplement 1**. GCs infected by rv GFP Bcl2-ires-cre in ROSA-lacZ reporter mice.

spine density did not exhibit significant differences between each pair (p > 0.09, Kolmogorov–Smirnov test; *Figure 3O*). We noted that, in comparison with WT GFP-expressing cells, Bcl2-expressing GCs did not show dramatically increased density (GFP: 0.028 ± 0.004, cre: 0.053 ± 0.011, Bcl2: 0.032 ± 0.007, p = 0.0092 cre vs GFP) or percentage (GFP: 1.40 ± 0.21, cre: 4.76 ± 0.66, Bcl2: 2.13 ± 0.39, p < 0.0001 cre vs GFP) of mushroom-like spines as naturally surviving NMDAR KO cells (*Figure 3P–Q*). However, Bcl2-rescued group displayed a strong trend of mushroom spine enhancement in *Grin1^{f/f}* mice (Bcl2 vs GFP: p = 0.08, *Figure 3Q*), but not in WT mice (Bcl2 vs GFP: p = 0.5, *Figure 3L*), suggesting that NR1 KO GCs accounted for a portion of neurons rescued by Bcl2 and they had more mushroom spines than those dying of reasons other than genetic ablation of NMDARs. Taken together, these data support that NMDAR was critical for spine outgrowth and that newborn GCs destined to die without the protection of Bcl2 were defective in spine formation.

## Mushroom spine density is increased in mature neurons upon deletion of NR1

Since Bcl2-rescued GCs exhibited a total spine reduction, but not a mushroom spine enhancement (*Figure 3H–Q*), it seems unlikely that the increases in spine volume and AMPA currents associated with NMDAR loss (*Figure 1I–L*) simply represent a homeostatic change in response to the decrease of total synaptic drive. However, two scenarios may explain the increase in mushroom spines found in surviving NR1 KO cells (*Figure 1*). The alteration of spine head diameter could be a direct consequence of NMDAR loss. Alternatively, the survival selection was biased towards neurons that happened to bear lots of mushroom spines or mature synapses. To test these possibilities, we developed a cre-ER retrovirus in which the expression of GFP and the inducible Cre recombinase creER^{t2} was bridged by the 2A sequence from Picornavirus (*Szymczak et al., 2004*) (rv CGS-creER; *Figure 4—figure supplement 1*) to induce NMDAR deletion in newborn GCs after their critical time window for survival. We injected rv CGS-creER or control rv CAG-GFP into *Grin1^{f/f}* and *Grin1^{f/+}* mice, respectively, and injected oil or tamoxifen (180 mg/kg, daily for 4 days) into the animals at 28 dpi. Mouse brains were collected at 56 dpi (28 days after the induction with tamoxifen) for morphological analyses. We first quantified GFP+ cell number from *Grin1^{f/f}* mice injected with rv CGS-creER and found no significant difference between oil and tamoxifen treatment (oil: 84 ± 48, n = 4 mice, tamo: 117 ± 15, n = 4 mice, p = 0.54; *Figure 4A–C*). Therefore, NMDAR activity was not required for the survival of newborn GCs that were 4 weeks old or older. Although there was no statistically significant decrease in total spine density in NMDAR KO cells (oil/creER: 2.20 ± 0.10, n = 20 frames, tamo/creER: 1.91 ± 0.12, n = 14 frames, tamo/GFP: 2.29 ± 0.11, n = 21 frames; *Figure 4D,E*), there appeared to be a strong trend of increased mushroom spine density (oil/creER: 0.133 ± 0.016, tamo/creER: 0.188 ± 0.027, tamo/GFP: 0.131 ± 0.015, p = 0.054 comparing tamo/creER with tamo/GFP; *Figure 4D,F*). In particular, the percentage of mushroom-shaped spines was markedly increased by 62% (oil/creER: 6.38 ± 0.91, tamo/creER: 10.35 ± 1.68, tamo/GFP: 6.17 ± 0.92, p = 0.024 comparing tamo/creER with tamo/GFP; *Figure 4G*). These data indicate that enhanced spine maturation was unlikely to be a compensatory effect from the stress of NMDAR-dependent cell survival.

Since NR1 KO newborn GCs display increased mushroom spine density at both immature and relatively mature stages during GC development, we next examined whether NMDARs functioned similarly in GCs generated during embryonic development. To delete NR1 in mature GCs, we engineered a lentivirus CAG-GFP-ires-cre (lv GFP-ires-cre; *Figure 5—figure supplement 1*) and injected the lv GFP-ires-cre virus or a control lv GFP virus into 8-week-old *Grin1^{f/+}* and *Grin1^{f/f}* mice. Lv-transduced GCs were analyzed at 28 dpi. A brief visual inspection of the sections revealed no obvious difference in the number of lv GFP-ires-cre-transduced cells in f/+ and f/f cells, consistent with the previous observation that NMDAR was not required for the survival of mature GCs (*Figure 5A,B*). Electrophysiological recordings showed that Cre-targeted cells did not respond to perforant path stimulation in the presence of the AMPAR blocker DNQX, confirming that Cre-expressing cells had no functional NMDARs (*Figure 5C,D*). There was no difference in total spine density between WT and NMDAR KO cells (*Grin1^{f/+}* GFP: 2.97 ± 0.14, n = 29 frames, *Grin1^{f/+}* cre: 3.09 ± 0.18, n = 28 frames, *Grin1^{f/f}* GFP: 3.15 ± 0.10, n = 71 frames, *Grin1^{f/f}* cre: 2.97 ± 0.08, n = 75 frames; *Figure 5E,F*). However, mature NMDAR KO GCs displayed significantly more mushroom spines than any other group (*Grin1^{f/+}* 0.128 ± 0.018, *Grin1^{f/+}* cre: 0.089 ± 0.013, *Grin1^{f/f}* GFP: 0.117 ± 0.009, *Grin1^{f/f}* cre: 0.193 ± 0.010, p < 0.0001; *Figure 5G*). Correspondingly, both the amplitude and the frequency of

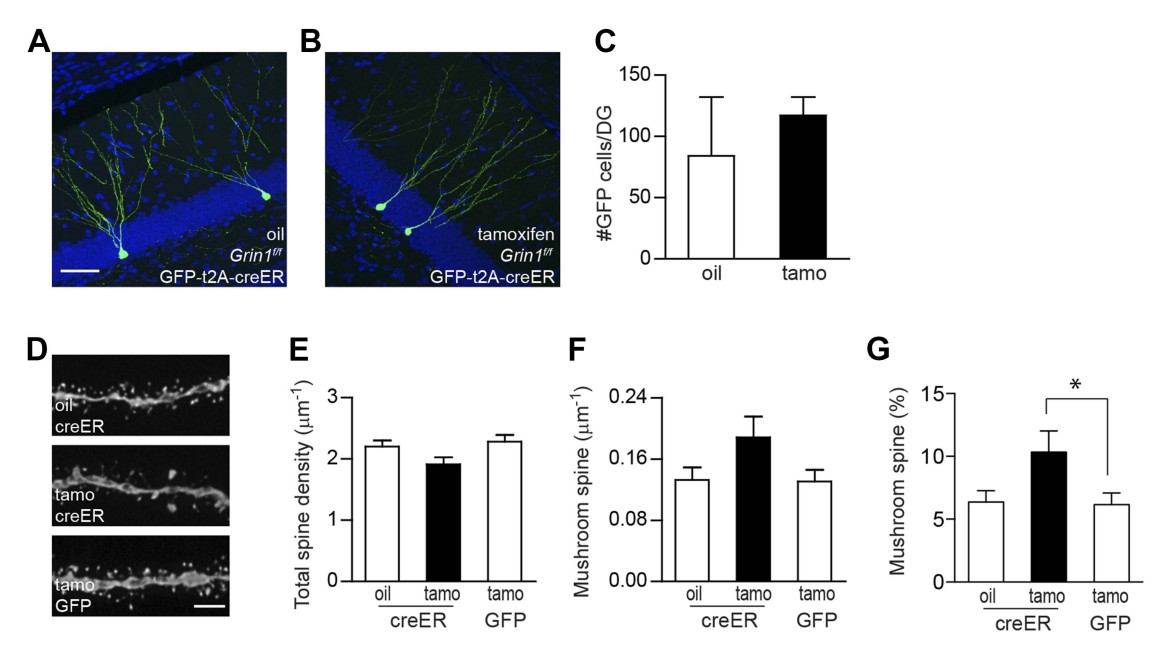

**Figure 4**. Using inducible cre to bypass the critical NMDAR-dependent cell survival. (**A**, **B**) Representative images of newborn GCs in *Grin1^f/f^* mice that were infused with rv CAG GFP-t2A-creER and treated with oil (**A**) or tamoxifen (**B**). (**C**) Deletion of the *Grin1* gene initiated in 4-week-old newborn GCs did not affect cell survival. (**D**) Representative images of dendritic processes in the outer molecular layer of rv GFP-t2A-creER-targeted newborn GCs treated with oil and tamoxifen, and of rv GFP-targeted newborn GCs treated with tamoxifen. (**E**–**G**) Quantification of total spine density (**E**), mushroom spine density (**F**) and mushroom spine percentage (**G**) in rv GFP-t2A-creER- and rv GFP-targeted newborn GCs (*p < 0.05). Scale bars: 50 µm (**A**, **B**) and 5 µm (**D**).

The following figure supplement is available for figure 4:

**Figure supplement 1**. The recombination efficiency of rv CAG GFP-t2A-creER was tested in the ROSA-lacZ mice.

AMPAR-mediated mEPSCs were increased in NMDAR KO mature cells (WT: n = 7 cells, KO: n = 6 cells, p < 0.01, Kolmogorov–Smirnov test; *Figure 5H–K*). These data demonstrated that spine maturation and synaptic AMPAR activity were enhanced in mature GCs in the absence of NMDAR.

Although adult-born GCs only constitute a small population of GCs in the DG and most lentivirus-labeled cells should be mature GCs, we could not completely rule out the possibility that our conclusions about NMDAR-dependent inhibition on spine maturation and AMPAR activity in mature GCs might be confounded by the small population of newborn cells possibly included in our analyses. To resolve this concern, and also to determine whether NMDAR-mediated inhibition of spine maturation was specific to GCs, we examined lv GFP-ires-cre-targeted CA1 pyramidal cells in *Grin1^f/f^* mice at 28 dpi (*Figure 6A,B*). Since no neurogenesis occurs in the CA1 area, the pyramidal cells we examined in this area should be exclusively mature cells. Images of the apical and basal dendrites of CA1 pyramidal cells were obtained from stratum lacunosum-moleculare and stratum oriens, respectively, and were analyzed separately. NMDAR KO pyramidal cells displayed a decreased total spine density in the apical dendrites but not the basal dendrites (apical GFP: 2.63 ± 0.12, n = 38 frames, cre: 2.12 ± 0.17, n = 15 frames, p = 0.023, basal GFP: 2.59 ± 0.11, n = 28 frames, cre: 2.44 ± 0.15, n = 15 frames, p = 0.44; *Figure 6C*). Hippocampal CA1 pyramidal neurons have distinct compartments with differential inputs, plasticity characteristics and mechanisms crucial for integrative function (*Spruston, 2008*). It is possible that NMDAR subunit composition, distribution and associated signaling pathways are different in apical and basal domains. Therefore, the spine turnover rate may be low and not sufficient to amount to a visible difference in total spine number in basal but not apical dendritic branches. For example, 96% of spines in the adult mouse visual cortex remained stable through weeks of live imaging (*Grutzendler et al., 2002*). Consistent with the data in lentivirus-transduced GCs, mushroom spine density was significantly higher in both apical and basal

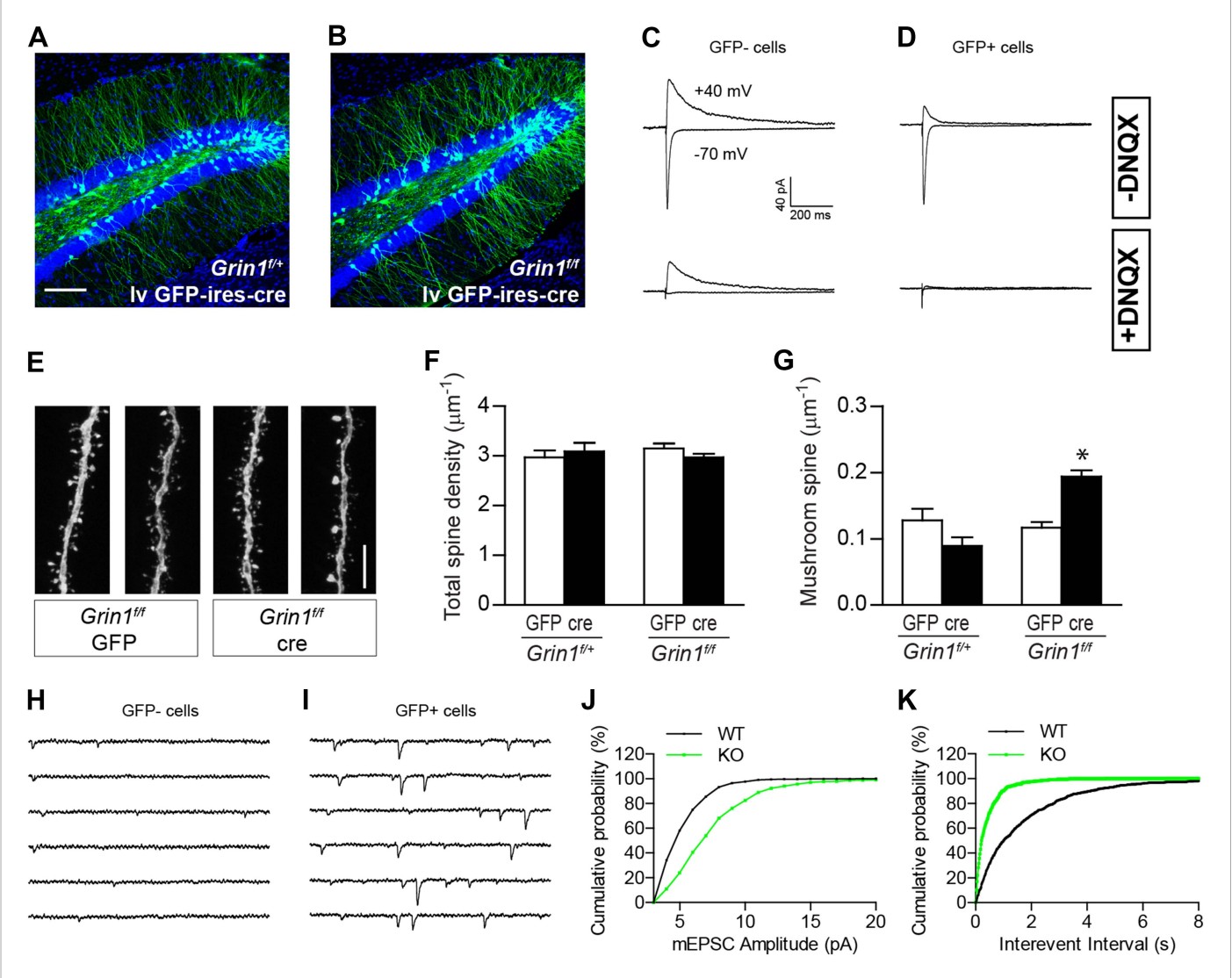

Figure 5. The effect of NR1 KO in mature GCs. (A, B) Representative images of mature GCs in *Grin1^{f/+}* (A) and *Grin1^{f/f}* (B) mice targeted by lv CAG-GFP-ires-cre. (C, D) Electrophysiological recordings of mature GCs in lv CAG-GFP-ires-cre-targeted mice. GFP− and GFP+ cells represent NR1 WT and KO GCs, respectively. (E) Representative images of dendritic processes in the outer molecular layer of GFP+ cells targeted by lv CAG-GFP and GFP-ires-cre. (F) NR1 KO mature GCs display similar total spine density as wild type GCs. (G) Mushroom spine density was increased in NR1 KO mature GCs (*p < 0.0001). (H, I) Sample traces of AMPAR-mediated mEPSCs in GFP− and GFP+ mature GCs in *Grin1^{f/f}* mice targeted by lv CAG GFP-ires-cre. (J, K) Cumulative plots of mEPSC amplitude (J) and frequency (K) confirmed that AMPAR-mediated activity was enhanced in NR1 KO mature GCs. Scale bars: 100 μm (A, B) and 5 μm (E).

The following figure supplement is available for figure 5:

Figure supplement 1. The recombination efficiency of lv CAG GFP-ires-cre was tested in ROSA-lacZ reporter mice 4 weeks after virus delivery.

dendrites of NMDAR KO pyramidal cells (apical GFP: 0.105 ± 0.012, cre: 0.192 ± 0.034, p = 0.0037, basal GFP: 0.050 ± 0.012, cre: 0.125 ± 0.030, p = 0.0077; *Figure 6D*). Therefore, spine maturation was also enhanced in CA1 pyramidal cells in the absence of NMDAR.

## Discussion

Using retro- and lentiviral vectors that encode the Cre recombinase, we deleted the *Grin1* gene in newborn GCs or mature neurons (GCs and CA1 pyramidal cells) to examine the impact of NMDAR

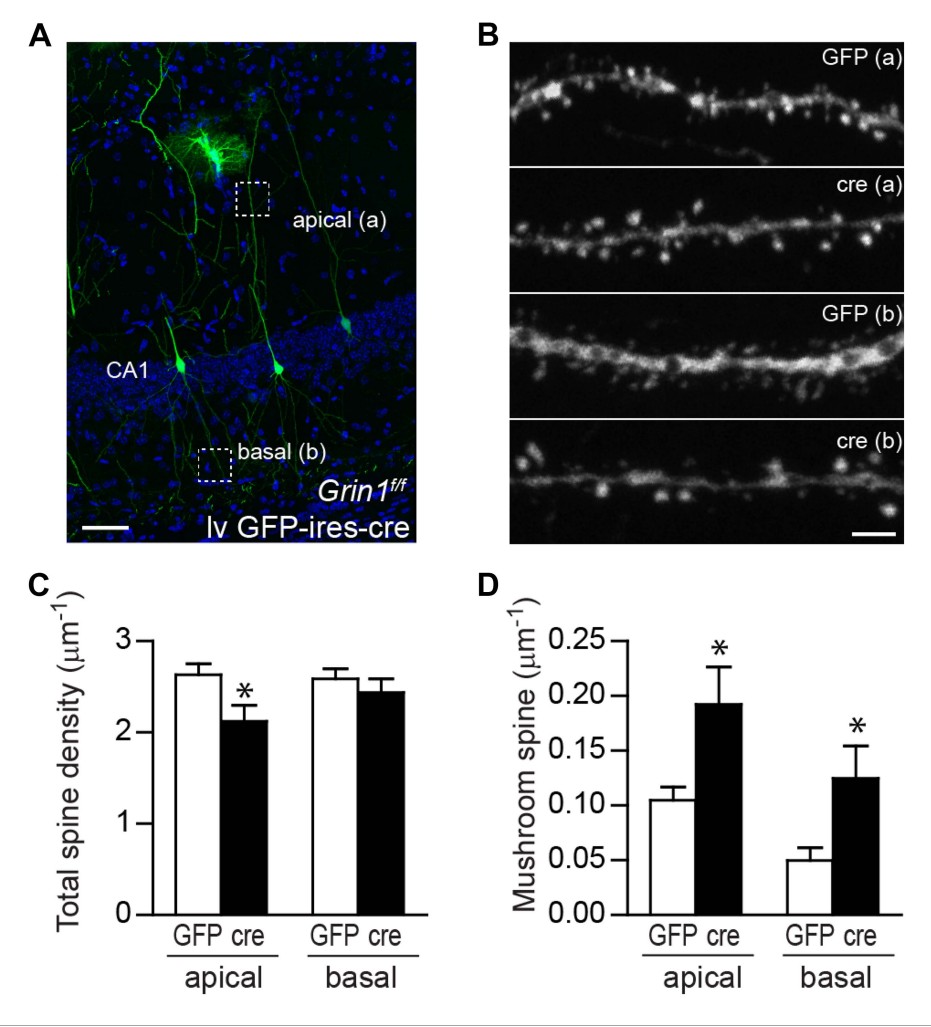

**Figure 6**. Mushroom spine density was increased in CA1 pyramidal cells in response to NR1 deletion in adult mice. (**A**) Representative image of CA1 pyramidal cells labeled by lv GFP-ires-cre. Dotted boxes indicate typical apical and basal dendrite segments for spine analysis. (**B**) Representative images of apical (**A**) and basal (**B**) dendrites in wild type and NR1 KO CA1 pyramidal cells. (**C**) Total spine density was decreased in the apical but not basal dendrites in CA1 pyramidal cells in response to NR1 deletion (*p < 0.05). (**D**) Mushroom spine density was increased in both apical and basal dendrites in NR1 KO pyramidal cells (*p < 0.01). Scale bars: 50 μm (**A**) and 2 μm (**B**).

loss on spine morphology. We found that NMDAR activity was required for initial spinogenesis and that a low level of spine density correlated with a defect in cell survival during the 3–4 weeks after new GCs were born, although NMDAR signaling did not appear to have a major impact on total spine numbers in mature neurons, whether they were neonate- or adult-generated. However, NMDAR was not critical for the maturation of dendritic spines once spines were formed. On the contrary, the density of mushroom spines was increased in NR1 KO GCs, accompanied by increased synaptic AMPAR activity. The latter phenomenon was observed not only in newborn GCs but also in mature GCs and CA1 pyramidal cells. Taken together, our data suggest that NMDARs serve to regulate the genesis of dendritic spines in developing neurons. Furthermore, NMDARs restrict synapse maturation and control spine morphology after spines have been formed, independently of the developmental stage.

We reported previously that NMDAR was required for the survival of newborn GCs in the adult mouse hippocampus (*Tashiro et al., 2006a*). Through detailed analyses, we found that NMDAR KO newborn GCs that were forced to survive with the expression of the pro-survival gene Bcl2 had

defects in spine growth and synaptogenesis. Because the time window of initial spine growth and synapse formation correlated with that of NMDAR-dependent cell survival, we deduced from these observations that the failure of synapse formation may be the underlying mechanism of cell death in NMDAR KO newborn GCs. In addition, we analyzed newborn GCs forced to survive through Bcl2 expression in WT mice and found that the overall population had lower total spine density compared to the cells that were selected to survive under natural conditions. This finding suggests that natural death of WT adult-born GCs might also be a result of the failure of spine formation in certain cell populations. Both voluntary exercise and enriched environment (EE) housing increased cell survival (*van Praag et al., 1999*; *Tashiro et al., 2007*; *Muotri et al., 2009*; *Zhao et al., 2014*). However, unlike the forced survival through Bcl2 expression, increased survival of GCs through physical exercise or EE did not lower the average spine density of newborn GCs (*Zhao et al., 2014*), suggesting that exercise and EE likely promote cell survival through enhancing spine formation in newborn GCs.

Many of the functional properties of NMDARs are highly sensitive to their subunit composition. Early expression of NR2A in organotypic hippocampal slices reduces the number of synapses and the volume and dynamics of spines, whereas overexpression of NR2B increases spine motility, indicating that the ratio of NR2B over NR2A controls spine motility and synaptogenesis (*Gambrill and Barria, 2011*). Furthermore, overexpression of NR3A reduces spine density over time by increasing spine elimination and decreasing spine stability (*Kehoe et al., 2014*). In adult-born GCs, there is a switch in synaptic NMDAR subunit composition from predominantly NR2B to NR2A during their development (*Ge et al., 2007*). Although the expression pattern of NR3A is unclear in nascent GCs, it is reasonable to speculate that NMDAR may be required for spine stabilization in addition to spine formation, depending on developmentally regulated synaptic expression of NMDARs containing a specific type of subunit. Alternatively, NMDARs are located not only at synapses but also at extrasynaptic sites. The roles played by extrasynaptic NMDARs are generally still elusive. It has been shown that filopodia-like protrusions or spines appear de novo after exogenous glutamate application or LTP induction (*Engert and Bonhoeffer, 1999*; *Maletic-Savatic et al., 1999*), raising a possibility that extrasynaptic NMDARs participate in the regulation of new spine formation. Given that newborn GCs display high levels of NMDARs before the formation of glutamatergic synapses (*Schmidt-Salzmann et al., 2014*), it is possible that the distinct roles of NMDARs during different stages of GC development may be attributed to different locations of NMDARs. Further study is needed to find out whether NMDAR location or subunit composition, or both, could be the determining factor of NMDAR's functional diversity.

NR1 deletion leads to decreased numbers of total spines and increased mushroom spines in developing cortical neurons (*Ultanir et al., 2007*). Here we revealed a very similar impact of NR1 loss on neurons born in the adult hippocampus and have provided additional evidence that adult neurogenesis follows a pattern resembling early development. However, the observation that NMDAR KO newborn GCs, irrespective of their age, displayed higher mushroom spine density and more synaptic AMPAR activity was unexpected. It contradicted the notion that NMDAR activity is required for spine enlargement and AMPAR recruitment (*Matsuzaki et al., 2004*; *Wang and Kriegstein, 2008*) and cannot be interpreted simply by differences between developing and mature synapses as previously speculated (*Ultanir et al., 2007*). Several possible mechanisms could account for enhanced spine maturation in the absence of NMDAR. First, the surviving cell population could be selected to survive because those cells have more mushroom spines or functional synapses. This possibility was ruled out because we observed a similar phenomenon in newborn GCs when NR1 was deleted at a later stage, after the NMDAR-dependent critical time window for cell survival. Furthermore, mature GCs and CA1 pyramidal cells also displayed increased mushroom spine density and synaptic activity in the absence of NMDAR, supporting our conclusion that enhanced synapse maturation is not a compensatory effect of NMDAR-dependent cell survival in newborn GCs. Alternatively, elevated mushroom spines could be potentially explained by a synaptic scaling mechanism, namely, NR1 KO cells might increase AMPAR activity in compensation for the loss of NMDAR activity. Indeed, we observed that AMPAR activity was significantly enhanced in both frequency and amplitude in the absence of the functional NMDAR. However, it has been reported that NMDAR activity itself is required for synaptic scaling (*Turrigiano et al., 1998*; *Wang et al., 2011*). Therefore, it is unlikely that enhanced spine maturation was a result of synaptic scaling. In an effort to resolve the discrepancy between our current observation and the traditional notion that NMDAR activity is required for AMPAR recruitment and spine enlargement, we tried to compare the systems used in our studies and others. Among the many differences between study systems, we found that

one difference could potentially account for the different observations. In studies that indicated an inhibitory role for NMDAR on spine maturation and AMPAR recruitment, NMDAR activity was lost due to the deletion of the *Grin1* gene (*Ultanir et al., 2007*; *Adesnik et al., 2008*), whereas in studies that supported NMDAR-dependent spine enlargement and AMPAR recruitment, NMDAR activity was blocked by antagonists (*Engert and Bonhoeffer, 1999*; *Maletic-Savatic et al., 1999*; *Zhu et al., 2000*). It was previously pointed out that a genetic approach might offer certain advantages over pharmacological manipulations (*Ultanir et al., 2007*). Here we propose that NMDAR in its inactive form inhibits spine maturation and AMPAR recruitment. The activation of NMDAR by binding of the neurotransmitter glutamate reverses its inhibitory function and allows the recruitment of AMPAR and spine enlargement. When NMDAR activity was blocked by antagonists, NMDAR was kept in its inactive state and thus prevented spine enlargement and AMPAR recruitment. However, when NMDAR was absent due to the loss of its essential subunit NR1, there was no longer inhibition from the inactive NMDAR, which resulted in uncontrolled spine enlargement and AMPAR recruitment. This hypothesis is in line with the critical role of NMDAR in LTP and reconciles the discrepancies observed when different methods were used to modulate NMDAR activity. Another possibility—not mutually exclusive—is that lack of NMDARs triggers expression of calcium-permeable AMPARs in newborn neurons and these receptors confer upon dendritic spines the ability to potentiate and to grow in the absence of NMDARs; this hypothesis awaits experimental clarification.

Experience constructs the neural network in the form of activity-dependent spine morphogenesis, which is central to memory formation and other adaptive changes of the brain. Although the current study has been mainly focused on newborn neurons in the adult brain, our findings suggest that the major neuronal activity detector, NMDAR, regulates the connectivity of specific neurons by common mechanisms in developing and mature nervous systems, including facilitation of new spine formation and control of the pace of existing spine maturation. By preventing precocious synapse maturation, NMDAR participates in 'neoteny', or the extension of the immature state of the brain, which is critical for subsequent and more complete maturation.

## Materials and methods

### Mice

*Grin1* floxed mice (*Tsien et al., 1996*) and the ROSA-lacZ reporter mice (B6;129S4-Gt(ROSA) 26Sor^tm1sho) were maintained as homozygous in the Salk mouse facility. For some of the experiments, $Grin1^{f/f}$ mice were bred to $Grin1^{f/+}$ mice so that littermate $Grin1^{f/+}$ mice were used as controls. Mice aged 5–7 weeks were used to examine the role of NR1 in newborn GCs, and those 8 weeks old or older were used to examine the role of NR1 in mature neurons. C57Bl/6 female mice, 6 weeks old, were used to test the retrovirus GFPBcl2-ires-cre. The animal protocols were all approved by the Salk Institutional Animal Care and Use Committee.

### Viruses

To examine stage-specific roles of the NMDAR, we developed four new cre-expressing viruses. Firstly, retrovirus CAG-GFP ires cre (rv GFP-ires-cre) was constructed for the purpose of gene deletion and the simultaneous visualization of the newborn cell morphology (*Figure 1—figure supplement 1*, 28 dpi). In the rv GFP-ires-cre vector, the Cre cDNA was placed after the internal ribosomal entry site to minimize the expression of Cre and potentially reduce cre-associated vector recombination in bacteria. When tested in the ROSA-lacZ reporter mice, rv GFP-ires-cre induced recombination in 97% of GFP+ cells at 6 dpi and in all GFP+ cells at 14 and 28 dpi. Secondly, a retrovirus CAG-GFPBcl2-ires-cre (rv GFPBcl2-ires-cre) was constructed so that the Cre-targeted cells also expressed the fusion protein of GFP and Bcl2 (*Figure 3—figure supplement 1*, 14 dpi). This virus allowed the deletion of the *Grin1* gene in newborn GCs and the expression of the pro-survival protein Bcl2 in the same cells so that NR1 KO cells would be prevented from going through apoptosis. The recombination efficiency of rv GFPBcl2-ires-cre was 100% at 7 dpi. Thirdly, an inducible retrovirus CAG-GFP-t2A-creER (rv GFP-t2A-creER) was generated so that deletion of the *Grin1* gene could be initiated at the desired age to bypass NR1-depedent cell survival (*Figure 4—figure supplement 1*, induction at 14 dpi). To test the recombination efficiency of rv GFP-t2A-creER virus in vivo, we gave tamoxifen (180 mg/kg, daily for 4–5 days) to virus-injected ROSA26-lacZ reporter mice starting from 14 and 28 days after virus injection. At 14 dpi, we observed 16% background recombination with oil control and 85%

**Table 1.** Total number of spines evaluated in *Figure 1G*

| Spine classes | WT | KO |
|---|---|---|
| Stubby | 742 | 391 |
| Mushroom | 44 | 97 |
| Thin | 3884 | 1643 |
| Filopodia | 202 | 121 |

recombination in GFP+ cells. At 28 dpi, recombination efficiency was 97%. Lastly, a Cre-expressing lentivirus CAG-GFP-ires-cre was developed for deletion of the *Grin1* gene in both mature and newborn cells to examine the function of NR1 in mature neurons (*Figure 5—figure supplement 1*, lv GFP-ires-cre, 29 dpi). Because mature cells greatly outnumbered immature cells, the Cre-targeted cells largely represented mature cells. The recombination efficiency of the lv CAG-ires-cre in ROSA reporter mice was 85% at 29 dpi. Recombinant retroviruses and lentiviruses were prepared in 293T cells as described before (*Zhao et al., 2006*; *Tashiro et al., 2006b*).

## Immunohistochemistry and cell quantification

Mouse brain sections of 40 μM thickness were prepared with a sliding microtome as described in detail (*Zhao et al., 2006*). Brain sections of one-in-six series were selected for DAPI staining. GFP+ cells in the GC layer were visualized and counted manually with a Nikon E800 microscope (Melville, NY, United States). The total number of labeled GFP+ cells per DG was then estimated by multiplying the number by 6.

## Confocal imaging and spine analysis

All images were acquired through the Bio-Rad R2100 confocal system (Berkeley, CA, United States) or the Zeiss 710/780 confocal system (Germany). Images of the whole cell morphology of GFP+ cells were taken with a 40× objective (Bio-Rad R2100) or a 25× W objective (Zeiss 710/780). GFP+ cells with at least one dendritic process terminating at the outer molecular layer were randomly picked for imaging (every nth cell depending on the labeling efficiency). If the number of labeled cells from the one-in-six series was too low to allow for 5–10 cells to be imaged, more sections were sampled to obtain enough cells from an individual mouse or until all sections were used. For spine analyses, dendritic processes of GFP+ cells in the outer molecular layer were imaged with a 60× oil objective (NA 1.4, Nikon, on Bio-Rad R2100) or with a 63× oil objective (NA 1.4, Zeiss, on Zeiss 710). The raw confocal image files were subjected to 10 iterations of deconvolution (AutoDeblur, AutoQuant, Troy, NY, United States). Dendrite measuring and spine analyses have been described in detail before (*Zhao et al., 2006*). The categorization of dendritic spine shape was based on qualitative criteria (*Harris et al., 1992*; *Parnass et al., 2000*). For classification of mushroom spines, major and minor axes of each spine head were measured with NeuronStudio program. A spine was judged to be of mushroom type if the head area (estimated with the function $\frac{1}{4} \times \pi \times D_{major} \times D_{minor}$) was ≥0.4 μm². The absolute numbers of each spine type for the NR1 KO experiment (*Figure 1G*) and the Bcl2 rescue experiment (*Figure 3L,Q*) are summarized in *Tables 1–3*.

## Electrophysiology

Electrophysiological recordings of NR1 WT and KO cells were performed using a protocol that was described in detail in a previous study (*Mu et al., 2011*). Specifically, mice injected with retrovirus- or lentivirus-expressing GFP-ires-cre were anesthetized by isoflurane inhalation. Mouse brains were immediately removed and placed in ice-cold dissection buffer (in mM choline chloride 110, KCl 2.5, NaH$_2$PO$_4$ 1.3, NaHCO$_3$ 25.0, CaCl$_2$ 0.5, MgCl$_2$ 7, glucose 20, Na-ascorbate 1.3, and Na-pyruvate 0.6).

Horizontal slices (200 μm thick) were prepared using a Leica VT1000S vibrotome (Germany) and incubated for at least 1 hr at room temperature before recording in standard ACSF (in mM NaCl 125, KCl 2.5, NaH$_2$PO$_4$ 1.3, NaHCO$_3$ 25, CaCl$_2$ 2, MgCl$_2$ 1.3, Na-ascorbate 1.3, Na-pyruvate 0.6, and glucose 10) that was saturated with 95% O$_2$ and 5% CO$_2$. Whole-cell perforated patch recordings were obtained from GCs visualized using an upright microscope (BX51WI; Olympus) with infrared differential interference contrast optics (Japan).

**Table 2.** Total number of spines evaluated in *Figure 3L*

| Spine classes | GFP | cre | Bcl2 |
|---|---|---|---|
| Stubby | 306 | 164 | 248 |
| Mushroom | 23 | 12 | 24 |
| Thin | 2545 | 1635 | 1835 |
| Filopodia | 65 | 59 | 104 |

**Table 3**. Total number of spines evaluated in *Figure 3Q*

| Spine classes | GFP | cre | Bcl2 |
| --- | --- | --- | --- |
| Stubby | 572 | 208 | 222 |
| Mushroom | 101 | 108 | 60 |
| Thin | 6901 | 1984 | 2342 |
| Filopodia | 160 | 78 | 164 |

The Cre-targeted cells were visually identified by their green fluorescence. The following drugs were used for blocking certain activities: 50–100 µM picrotoxin to block GABAergic synaptic transmission, 10 µM DNQX to block AMPAR-mediated activity, 25 µM APV to clock NMDAR-mediated activity and 1 µM TTX to block action potentials. All experiments were performed at room temperature. A bipolar tungsten electrode was used for extracellular stimulation of the perforant path, and GCs were held at −70 mV in voltage-clamp mode unless stated otherwise.

## Electron microscopy and image analysis

Tissue processing for electron microscopy was performed as described previously (*Toni et al., 2008*). Briefly, mice were transcardially perfused with 4% paraformaldehyde and brains were cut coronally at a thickness of 50 µm. Sections were cryoprotected, briefly freeze-thawed four times in liquid nitrogen and treated with 0.3% hydrogen peroxide. After a block with 0.5% bovine serum albumin, slices were incubated 40 hr with rat anti-GFP antibody (1:500, Chemicon) at 4°C on a shaker. After washing, sections were incubated for 5 hr at 5°C in biotinylated secondary antibody (goat anti-rabbit, Fac fragment, 1:500, Chemicon). Sections were then incubated in avidin biotin peroxidase complex (ABC Elite, Vector laboratories), followed by 3,3′-diaminobenzidine tetrachloride for 10–20 min to obtain a dark residue in labeled cells. Sections were next postfixed in 2.5% glutaraldehyde, followed by 1% osmium tetroxide, dehydrated in ascending concentrations of ethanol and then acetone, and finally embedded in Epoxy resin. Sections of a thickness of 50 nm were contrasted with uranyl acetate followed by lead citrate and observed on a Philips CM10 electron microscope (Hillsboro, OR, United States), at a magnification of 13,500×. Synapses were defined by the presence of at least three presynaptic vesicles within 50 nm of the presynaptic membrane, a clearly defined synaptic cleft and a postsynaptic density. Spines were serially sectioned and the surface area of each segment was measured on every image. Volumes were obtained by multiplying the surface area, the section thickness and the number of sections.

## Statistical analysis

All data were presented as mean $\pm$ standard error. Statistic comparisons were done using unpaired two-tailed t-test, except that the Kolmogorov–Smirnov test was used for data analyses on AMPAR mEPSCs.

## Acknowledgements

We are grateful to ML Gage for editorial comments, B Miller for virus preparation, E Mejia, I Chen and J Jou for help with histology and confocal imaging, and W Deng, JB Aimone and D Clemenson for discussions. We also thank J Fitzpatrick, J Kasuboski and Y Sigal at the Waitt Advanced Biophotonics Center for technical support, and the Electron Microscope Facility at the University of Lausanne for the electron micrography. This work was supported by the McDonnell Foundation, the JPB Foundation, Mathers Foundation, the Ellison Foundation and a grant from NIH to FH Gage. Y Mu is funded in part by National 1000-Young-Talent Program of China, the Fundamental Research Funds for the Central Universities, HUST: 2014TS059 and the National Natural Science Foundation of China (91332106). N Toni is funded by the Swiss National Science Foundation.

## Additional information

### Funding

| Funder | Grant reference | Author |
| --- | --- | --- |
| James S. McDonnell Foundation (JSMF) | | Fred H Gage |
| JPB Foundation | | Fred H Gage |

| Funder | Grant reference | Author |
|---|---|---|
| G Harold and Leila Y. Mathers Foundation | | Fred H Gage |
| Ellison Medical Foundation (EMF) | | Fred H Gage |
| National Institutes of Health (NIH) | | Fred H Gage |
| National 1000-Young-Talent program of China | | Yangling Mu |
| National Natural Science Foundation of China (NSFC) | 91332106 | Yangling Mu |
| Fundamental Research Funds for the Central Universities, HUST | 2014TS059 | Yangling Mu |
| Schweizerischer Nationalfonds zur Förderung der Wissenschaftlichen Forschung | | Nicolas Toni |

The funders had no role in study design, data collection and interpretation, or the decision to submit the work for publication.

## Author contributions

YM, CZ, Conception and design, Acquisition of data, Analysis and interpretation of data, Drafting or revising the article; NT, Acquisition of data, Analysis and interpretation of data, Drafting or revising the article; JY, Acquisition of data, Drafting or revising the article; FHG, provided support, Conception and design, Drafting or revising the article

## Ethics

Animal experimentation: This study was performed according to the Salk Institute animal care and use protocol #09-060. The protocol was approved by the institutional animal care and use committee (IACUC) of the Salk Institute.

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
