## [Decision Letter]

Thank you for submitting your work entitled “Distinct Roles of NMDA Receptors at Different Stages of Granule Cell Development in the Adult Brain” for peer review at *eLife*. Your submission has been favorably evaluated by a Senior Editor, a Reviewing Editor, and two reviewers.

The reviewers have discussed the reviews with one another and the Reviewing editor has drafted this decision to help you prepare a revised submission. Both reviewers expressed interest in this manuscript but also felt that the manuscript needs to be improved before it can be published in *eLife*.

One reviewer wrote: “This study follows up on an extensive literature on the role of NMDA receptors in synaptic formation and plasticity. The authors used several mouse models and viral constructs to test whether lack of NMDA R1 affects spine formation of newborn neurons as well as their survival in the dentate gyrus. It is very well known that NMDA receptors are essential for spine formation both during embryonic development and throughout the adulthood, and the findings reported in this paper confirm that this is the case for adult-born neurons in the dentate gyrus. The novelty of the paper comes from the apparent role of NR1 in the survival of newborn neurons”. The other reviewer wrote: “The study by Mu, Zhao and colleagues reveals a role for NMDA receptors in the development of dentate gyrus neurons and the regulation of spine number and size. They find that loss of the NR1 subunit of the NMDA receptor in these neurons in the first 4 weeks of their development leads to a decreased number of spines and an increase density of mushroom spines. This is accompanied by an increase in mEPSC amplitude and frequency, indicating an increase in functional Glu synapses. I think this study is an interesting addition to the neurogenesis field, and the experiments are well executed and presented”.

Both reviewers have specific suggestions about how the manuscript can be improved:

1) The report can be substantially improved by providing far more detailed analysis, as the data already exist. There are several observations indicating that spine size, which can range over two orders of magnitude, is likely to be important for their function. Larger spines can greatly outlast small spines (months compared with hours); they contain large synapses with more AMPA receptors and hence are functionally stronger than small spines. It appears that insertion of GluR1 is the key link that maintains the balance between synaptic strength and spine size. With this already established knowledge, the authors could present their own study in a more informative way. For example, given the large heterogeneity of spine structure and more-or-less continuum from filopodia to thin to stubbed to mushroom spines, I would like the authors to define exactly what is included in “total spine density”; what is the proportion of the other types of spines (and not only the mushroom ones); what exactly they mean by the mushroom spine (neck size, length)? Can they provide the absolute numbers for each type, at least in the supplementary table? Can they calculate the volume of spines – if the overall volume is not significantly different, perhaps there is a re-distribution of the calcium entry which may affect the survival of neurons? For example, in the absence of spines, neurons can still maintain synapses made directly onto their dendritic shafts. Shaft synapses are likely to produce larger synaptic currents than spine synapses, leading to larger increase in intracellular calcium concentrations. Such inputs may lead to cell death. The authors have already done EM, but they report only synapses onto the mushroom spine heads. Are there any synapses on dendrites? In addition, it would be very informative to provide some data on the plasticity of spines – they have all the tools and time-lapse imaging could be accomplished.

2) In the Discussion, the authors try to reconcile their findings with already published ones, as it has been accepted that NMDARs are required for recruitment of AMPA receptors to transition the spine from silent to functionally active synapse. They cite differences in methodology (genetic vs. pharmacological manipulation) as a cause for discrepancy. But – their findings actually may be the foundation of a major discovery: that lack of NR1 may trigger expression of calcium permeable AMPA receptors on the mushroom spines of newborn neurons, thus ensuring that the given synapse is functional, i.e. that it can be potentiated regardless of the presence of NMDA receptor?

3) The primary finding that NR1 deletion leads to decreased total spines and increased mushroom spines is quite interesting. This was shown previously by Anirvan Ghosh's lab in cortical neuron development (Ultanir et al PNAS 2007). This similarity in findings should be discussed.

4) The Bcl2 expression experiment should be unpackaged and discussed more, as in its current form it is difficult to extract conclusions. First, in normal mice overexpression of Bcl2 leads to a decrease in dendritic length and spine density. Why would this be? Is this a consequence of Bcl2 expression or due to activity dependent competition processes? Next, the authors show that cells with overexpression of Bcl2 and deletion of NR1 show reduced spine density without a change in mushroom spine number. The authors are making a link between NR1 deletion and mushroom spines, however, wouldn't one expect the NR1KO cells rescued from cell death should also show increased mushroom spines? Why does Bcl2 expression “rescue” the effect of NR1KO on mushroom spines?

5) By the way the Bcl2 experiments are presented and discussed, it remains unclear whether spine formation is required for the survival of adult born GCs. In the Discussion, the authors state: “this finding suggests that natural death of WT adult-born GCs might also be a result of the failure of spine formation in certain cell populations.” Its not clear to me as a reader how the forced survival of cells with less spines (Bcl2-overexpression, NR1KO) can lead to this conclusion.

6) The loss of NR1 from mature neurons leading to an increase in mushroom spines is an interesting finding. Can the authors comment on why this produces a decrease in apical spine density in mature CA1 neurons?

7) In the fourth paragraph of the Discussion the authors discuss their findings of enhanced spine maturation in NR1KO. This paragraph should include citations to statements made about previous studies (for example they argue differences in studies using pharmacological vs genetic manipulations without citations to the relevant literature). Also, this part of the discussion would be a good place to compare their findings to the Ultanir PNAS paper, as the Ultanir paper had a similar discussion as to how their findings of increased spine size/AMPA currents and decreased spine density compares to other pharmacological manipulations.

---

## [Author Response]

*1) The report can be substantially improved by providing far more detailed analysis, as the data already exist. There are several observations indicating that spine size, which can range over two orders of magnitude, is likely to be important for their function. Larger spines can greatly outlast small spines (months compared with hours); they contain large synapses with more AMPA receptors and hence are functionally stronger than small spines. It appears that insertion of GluR1 is the key link that maintains the balance between synaptic strength and spine size. With this already established knowledge, the authors could present their own study in a more informative way. For example, given the large heterogeneity of spine structure and more-or-less continuum from filopodia to thin to stubbed to mushroom spines, I would like the authors to define exactly what is included in “total spine density”; what is the proportion of the other types of spines (and not only the mushroom ones); what exactly they mean by the mushroom spine (neck size, length)? Can they provide the absolute numbers for each type, at least in the supplementary table? Can they calculate the volume of spines – if the overall volume is not significantly different, perhaps there is a re-distribution of the calcium entry which may affect the survival of neurons? For example, in the absence of spines, neurons can still maintain synapses made directly onto their dendritic shafts. Shaft synapses are likely to produce larger synaptic currents than spine synapses, leading to larger increase in intracellular calcium concentrations. Such inputs may lead to cell death. The authors have already done EM, but they report only synapses onto the mushroom spine heads. Are there any synapses on dendrites? In addition, it would be very informative to provide some data on the plasticity of spines– they have all the tools and time-lapse imaging could be accomplished*.

We appreciate the reviewers’ advice on how to present the study in a more informative way. We have reanalyzed the imaging data sets for the old Figures 1 and 2 and provided the proportions and absolute numbers for each spine type in new Figure 1, Figure 3, Figure 3 and Tables 1, 2 and 3, respectively. Correspondingly, we have explained clearly what was included in “total spine density” and how we defined a mushroom spine in the revised text. Thanks to the reviewer for pointing out the possibility that a re-distribution of calcium entry may affect the survival of neurons. We have checked our EM data more carefully and found symmetric, presumably GABAergic, synapses made directly onto dendritic shafts. Calculation of the total volume of spine heads was also performed on the data shown in Figure 1, which indicated a significant difference between WT and NR1 KO GCs. Since the spines included in volume measurement were randomly selected and comprised of mainly thin and mushroom spines, we cannot differentiate whether the volume change is owing to mushroom spine alone or distinct spine forms. Furthermore, the GABAergic synapses located on dendritic shafts could be excitatory in newborn GCs due to their high intracellular chloride concentration and allow Ca^2+^ influx. Therefore, we concluded that changes in the amount of Ca^2+^ influx and/or a re-distribution of Ca^2+^ entry may affect the survival of nascent neurons. As for the time-lapse imaging experiment suggested by the reviewers, we agree that data on the plasticity of spines would be very informative. However, LTP induction together with two-photon imaging has to be performed in the NR1 knockout neurons, which is technically difficult given the very low number of available cells and will take much longer than two months to set up and complete. In addition, while the reviewers indicate we have the equipment to do the live imaging, that was from a study years ago and we do not currently have that method working in the lab. Since this could be a subject of another study, we did not provide experimental results about spine plasticity in the present manuscript.

2) In the Discussion, the authors try to reconcile their findings with already published ones, as it has been accepted that NMDARs are required for recruitment of AMPA receptors to transition the spine from silent to functionally active synapse. They cite differences in methodology (genetic vs. pharmacological manipulation) as a cause for discrepancy. But – their findings actually may be the foundation of a major discovery: that lack of NR1 may trigger expression of calcium permeable AMPA receptors on the mushroom spines of newborn neurons, thus ensuring that the given synapse is functional, i.e. that it can be potentiated regardless of the presence of NMDA receptor?

We thank the reviewers for suggesting a very interesting hypothesis or an alternative explanation for our major discovery. This possibility has been added in our revised manuscript.

*3) The primary finding that NR1 deletion leads to decreased total spines and increased mushroom spines is quite interesting. This was shown previously by Anirvan Ghosh's lab in cortical neuron development (Ultanir et al PNAS 2007). This similarity in findings should be discussed*.

We thank the reviewers for suggesting a very interesting hypothesis or an alternative explanation for our major discovery. This possibility has been added in our revised manuscript.

*4) The Bcl2 expression experiment should be unpackaged and discussed more, as in its current form it is difficult to extract conclusions. First, in normal mice overexpression of Bcl2 leads to a decrease in dendritic length and spine density. Why would this be? Is this a consequence of Bcl2 expression or due to activity dependent competition processes? Next, the authors show that cells with overexpression of Bcl2 and deletion of NR1 show reduced spine density without a change in mushroom spine number*. *The authors are making a link between NR1 deletion and mushroom spines, however, wouldn't one expect the NR1KO cells rescued from cell death should also show increased mushroom spines? Why does Bcl2 expression “rescue” the effect of NR1KO on mushroom spines?*

We agree that the current form of Bcl2 experiment expression is not clear enough. The logic behind this set of experiment is: Bcl2 suppresses apoptosis and therefore rescues general cell death, including the death caused by NR1 loss (new Figure 3). Since the rescued cells exhibit a decrease in spine density as compared to those surviving naturally, it can be deducted that spine reduction may be a defect and Bcl2 overexpression cancels out this effect. We have substantially revised the result section to explain how we reached our conclusions, step by step.

*5) By the way the Bcl2 experiments are presented and discussed, it remains unclear whether spine formation is required for the survival of adult born GCs. In the Discussion, the authors state: “this finding suggests that natural death of WT adult-born GCs might also be a result of the failure of spine formation in certain cell populations.” Its not clear to me as a reader how the forced survival of cells with less spines (Bcl2-overexpression, NR1KO) can lead to this conclusion*.

Similar to last question, we have changed the way by which Bcl2 experiment is presented to address the reviewers’ concerns.

*6) The loss of NR1 from mature neurons leading to an increase in mushroom spines is an interesting finding*. *Can the authors comment on why this produces a decrease in apical spine density in mature CA1 neurons?*

We have provided some thoughts on this phenomenon in the revised Result section.

*7) In the fourth paragraph of the Discussion the authors discuss their findings of enhanced spine maturation in NR1KO. This paragraph should include citations to statements made about previous studies (for example they argue differences in studies using pharmacological vs genetic manipulations without citations to the relevant literature). Also, this part of the discussion would be a good place to compare their findings to the Ultanir PNAS paper, as the Ultanir paper had a similar discussion as to how their findings of increased spine size/AMPA currents and decreased spine density compares to other pharmacological manipulations*.

We appreciate the reviewer’s advice, and this part of Discussion has been revised as suggested.